# Patterns of Prescription Switching in a Uniform-Pricing System for Multi-Source Drugs: A Retrospective Population-Based Cohort Study

**DOI:** 10.3390/healthcare13182339

**Published:** 2025-09-17

**Authors:** Dong Han Kim, Song Hee Hong

**Affiliations:** 1College of Pharmacy, Seoul National University, Seoul 08826, Republic of Korea; kimdh830@snu.ac.kr; 2Research Institute of Pharmaceutical Science, Seoul National University, Seoul 08826, Republic of Korea

**Keywords:** generic substitution, bioequivalence, multi-source prescription switches, drug price competition, pharmacy benefit management

## Abstract

**Background:** Generic drugs account for approximately 40% of the Korean prescription drug market, despite limited generic substitution at the point of dispensing. This suggests that switching between originator and generic drugs often occurs at the point of prescription. Physicians, in fact, have opposed pharmacy-level substitution due to concerns about the clinical equivalence of generics, despite the regulatory confirmation of their bioequivalence. Importantly, multi-source prescription switching (MSPS) may reflect discretionary prescribing behavior, underscoring the need for targeted benefit policies to enhance substitutability and promote effective competition among multi-source drugs. This study aimed to quantify the extent of physician-initiated MSPS among adults with hypertension or diabetes and to identify factors associated with these switching behaviors. **Methods:** We conducted a retrospective cohort study using Korean National Health Insurance claims data. The studied cohort consisted of patients newly initiated, between January and June 2014, on a pharmaceutically equivalent and bioequivalent antihypertensive or antidiabetic drug. Patients were followed for up to 24 months to identify MSPS episodes occurring during drug therapy courses, which were defined as 12 ± 3 consecutive visits resulting in prescriptions for pharmaceutically equivalent, bioequivalent multi-source drugs. An MSPS episode was defined as a change in product code—uniquely identifying a multi-source drug—within the same pharmaceutically equivalent drug code between any two consecutive prescriptions within the course. We estimated the mean MSPS rate and assessed variation by patient characteristics, drug types, physician practices, and geographic regions. **Results:** Among 1,325,334 identified drug therapy courses, 17.06% involved at least one MSPS. Switching rates varied substantially (coefficient of variation = 227%) by physician practice setting (e.g., public health center branches: 26%; tertiary hospitals: 15%) and by drug market size (e.g., glimepiride: 29%; cilnidipine: 1%). In contrast, patient age and gender were not associated with switching behavior. **Conclusions:** In Korea, physicians frequently switch prescriptions between originator and generic drugs, even as generic substitution at the pharmacy level remains uncommon. The substantial variation in MSPS across provider settings and drug markets—but not by patient characteristics—underscores the need for targeted pharmacy benefit policies to promote effective substitutability and competition among multi-source drugs.

## 1. Introduction

In Korea, national pharmacy benefit policies have been implemented to encourage price competition among multi-source drugs. One such initiative provides financial incentives to pharmacies by returning 30% of the savings they achieve from purchasing drugs at prices lower than the reimbursement ceiling [1]. Despite these efforts, the prices of generic drugs have remained at or just below the ceiling price—set at 53.55% of the originator’s pre-entry price—indicating limited price competition in the market [2,3,4].

The failure of this incentive policy may lie in the limited authority of pharmacists to influence generic drug selection. Although pharmacists in Korea are legally permitted to make generic substitutions, they rarely exercise this right due to concerns about jeopardizing their relationships with prescribing physicians. As a result, the generic substitution rate at the pharmacy level remains extremely low—just 0.2% [1,5,6]. This raises the question: how did generics come to occupy nearly 40% of the Korean prescription drug market [7]? Evidently, such switching must have occurred at the point of prescription (POP), with physicians, not pharmacists, initiating the substitution.

The prevalence of multi-source prescription switching (MSPS) is difficult to justify, particularly in a regulatory environment where all interchangeable drugs are uniformly priced and required to demonstrate bioequivalence under MFDS standards [8,9,10,11]. For patients, such switching offers no financial benefit, as copayment levels remain constant regardless of the product dispensed. The rationale weakens further when clinical equivalence is contested—as some physicians have argued to oppose generic substitution at the point of dispensing—raising concerns about therapeutic consistency. Physicians similarly lack intrinsic incentives to engage in MSPS under a uniform pricing system, unless extrinsic motivations are involved. This dynamic has heightened concerns over manufacturer-driven inducements and led to the enforcement of anti-kickback regulations, which explicitly prohibit pharmaceutical companies from offering financial or non-financial incentives to influence prescribing behavior [12].

Generic switching in Korea differs markedly from practices observed in other developed countries, such as the United States (Table 1). In Korea, switching more commonly occurs between one generic drug and another, rather than from the originator to a generic version [13,14]. In contrast, U.S. generics do not have their own brands but international nonproprietary names (INNs) and are frequently substituted at the pharmacy level. Korean generics, however, are marketed under proprietary brand names and are often produced by reputable domestic pharmaceutical companies—many of which have historically introduced world-leading drugs into the Korean market through licensing agreements [2,15]. This allows physicians in Korea to prescribe a specific generic brand by name. Although pharmacists are legally permitted to substitute a bioequivalent product, they are required to notify the prescriber, creating an additional procedural barrier. Moreover, because most pharmacies in Korea are small, independently owned businesses that rely heavily on prescriptions from nearby clinics, pharmacists often refrain from exercising substitution rights to preserve favorable relationships with prescribers. This dynamic contributes to the extremely low substitution rate at the point of dispensing.

Korea’s pharmacy benefit policies have undergone multiple reforms since the introduction of mandatory National Health Insurance in 1989 [16]. Earlier policies employed differential pricing and reimbursement schemes, in which drug prices were set based on both the originator’s pre-entry price and the order in which generic competitors entered the market [17]. Specifically, after the entry of the first generic, the originator’s price was reduced to 80% of its original level. The first five generic entrants were reimbursed at 85% of the originator’s adjusted price, with subsequent generics facing additional stepwise reductions. Although this policy was intended to stimulate price competition and reward early generic entry, in practice, it failed to reduce drug prices meaningfully. Instead, it incentivized excessive promotional efforts by generic manufacturers to influence physicians’ prescribing behavior, as prices were fixed and prescribers had no financial motivation to select lower-priced alternatives. In response, the system was overhauled in 2012 and replaced with the current uniform pricing model, under which all multi-source products—originator and generics alike—are reimbursed at the same, reduced rate of 53.55% of the originator’s pre-entry price [2]. However, this reform also failed to curb promotional practices, as evidenced by multiple reports of illegal inducements offered to physicians in exchange for prescribing specific products [12,18].

Given this context, it is important to understand the role of physicians in multi-source drug markets where price signals are effectively absent. This study aims to examine the occurrence of multi-source prescription switching (MSPS)—defined as physician-initiated switching between bioequivalent drug products—within ongoing drug therapy courses involving pharmaceutically equivalent medications. Specifically, the objectives are (1) to quantify the extent to which physicians switch between sources (e.g., from originator to generic, or between generics) during defined courses of therapy among adults with hypertension or diabetes; and (2) to assess how MSPS rates vary according to patient characteristics, drug products, physician practice settings, and geographic regions.

Understanding the drivers of MSPS can inform future policy strategies aimed at enhancing substitutability among multi-source drugs, which holds substantial potential for improving system efficiency and reducing prescription drug expenditures within Korea’s national health system.

## 2. Methods

### 2.1. Study Design

This study employed a retrospective cohort design using Korean National Health Insurance claims data (Wonju Korea). The study population consisted of patients who, between 1 July and 31 December 2014, were newly initiated on a pharmaceutically equivalent drug with a bioequivalent designation. Patients were followed for up to 24 months, until 31 December 2016, to identify episodes of multi-source prescription switching (MSPS). To ensure that only new initiators were included, patients with any prescription record for a bioequivalent version of the index drug within the 12 months preceding the index date were excluded.

For eligible patients, a defined course of bioequivalent multi-source drug therapy was constructed and served as the unit of analysis. Each course was required to consist of 12 ± 3 prescription visits and an average days’ supply of at least 15 days per prescription, to reflect consistent, long-term therapy rather than short-term or irregular use. A drug therapy course of 12 ± 3 prescription visits likely approximates a one-year treatment period, given the average of 33.5 days per prescription for diabetic patients under the Korean health insurance system [19]. The observation period for this study spanned from 1 July 2013 to 30 June 2016 (Figure 1).

### 2.2. Study Population and Unit of Analysis

The study population included beneficiaries of the Korean National Health Insurance Service (NHIS) who received outpatient prescriptions for oral antihypertensive or antidiabetic drugs between 1 July 2013, and 30 June 2016. Patients with hypertension or diabetes were selected for this study because both are highly prevalent in older Korean adults, are commonly managed with multi-source chronic medications, and entail frequent prescribing—making them ideal for investigating multi-source prescription switching (MSPS). The unit of analysis was a defined one-year course of drug therapy involving pharmaceutically equivalent and bioequivalent multi-source drugs. Drug therapy courses were included only if they met the definition of 12 ± 3 prescription visits with an average days’ supply of ≥15 days per prescription.

Drug therapy courses could begin any time during the observation period, as long as the full set of visits occurred consecutively within the same drug code (i.e., the same active ingredient, dose, and formulation). Only drugs with at least four bioequivalent competitors were included, due to data availability restrictions from the Health Insurance Review and Assessment Service (HIRA), which withholds manufacturer-level prescription data for drug markets with fewer than four competitors to safeguard confidentiality.

### 2.3. Data Source

Data were obtained from the Health Insurance Review and Assessment Service (HIRA), which receives and adjudicates all health insurance claims submitted under Korea’s National Health Insurance system. This study used claims data spanning from 1 July 2013 to 30 June 2016. Upon exemption from the Institutional Review Board at Seoul National University, HIRA released all outpatient prescription data for oral antihypertensive and antidiabetic drugs prescribed between July and December 2014, provided that the drugs had at least four bioequivalent alternatives available at the time.

### 2.4. Independent Variables

#### 2.4.1. Number of Sources

The number of sources was defined as the number of distinct manufacturers marketing a bioequivalent product with the same active pharmaceutical ingredient (API), dose, and formulation. Each manufacturer is represented by a unique product code in the claims data. These codes serve as the operational basis for identifying bioequivalent products. In Korea, bioequivalence testing requirements are comparable to those of the U.S. FDA [20].

#### 2.4.2. Types of Physician Practice

In Korea, physicians may practice in a variety of settings with differing degrees of organizational control. Tertiary and general hospitals typically have stronger internal governance (e.g., P&T committees) and must meet requirements related to bed count, medical specialties, and residency training. Elderly care hospitals are largely privately owned and provide long-term inpatient care. Public sector facilities include public health centers (PHCs), PHC branches, and rural public healthcare facilities. PHC branches often employ physicians fulfilling mandatory military service, while rural public facilities may rely on nurses to serve medically underserved areas. Autonomy in prescribing is generally greater in public or smaller-scale settings, where formulary restrictions are minimal or absent.

#### 2.4.3. Insurance Type

Korea’s universal National Health Insurance was established in 1989, covering all citizens except specific groups such as veterans and the medically needy, who are covered by special programs. Patients under NHIS typically pay 30–50% of outpatient prescription costs, depending on the facility level: 30% for clinics, 40% for general hospitals, and 50% for tertiary hospitals. Patients under Medical Aid or Veterans programs face lower out-of-pocket burdens. This study categorized patients by insurance type: NHIS, Medical Aid, or Veteran.

### 2.5. Statistical Analysis

As this was a population-based study, inferential statistics were not applied. Instead, descriptive statistics—including population-level parameters such as the mean MSPS rate and standard deviation—were used to quantify the proportion of drug therapy courses in which an MSPS occurred. These parameters were further stratified and compared across patient demographics, drug characteristics, physician practice settings, and geographic regions to explore variation in MSPS patterns. Graphical tools were employed to visualize this variation and highlight outliers.

### 2.6. Results

#### 2.6.1. Description of Studied Drug Therapy Courses

A total of 1,325,334 drug therapy courses met the criteria for inclusion as defined courses of bioequivalent multi-source drug therapy for patients with hypertension or diabetes (Table 2). Because a single patient could initiate multiple courses depending on the number of different medications prescribed, the unit of analysis reflects individual therapy courses rather than individuals.

The gender distribution was relatively balanced, with a slightly higher proportion of courses attributed to female patients (51.24%). Age-wise, the largest proportion of courses was among patients in their 60s (30.58%), followed by those in their 70s (26.31%) and 50s (24.70%). Those under 50 had a share of 8.69%.

Most therapy courses (93.88%) were covered under Korea’s National Health Insurance, either through employment-based or community-based enrollment. The remaining courses were associated with patients receiving government-supported coverage, such as Medical Aid or Veterans programs.

Regarding the number of sources, more than half of the drug therapy courses (56.54%) involved drugs with 75 or more bioequivalent sources, reflecting highly competitive multi-source markets. In contrast, only 6.65% of the courses involved drugs with fewer than 25 sources, indicating limited competition in those therapeutic classes.

#### 2.6.2. Multi-Source Prescription Switching (MSPS) and Its Variation

The overall proportion of defined drug therapy courses that experienced at least one instance of MSPS was 17.06% (μ), with a standard deviation (σ) of 38%, resulting in a coefficient of variation (CV) of 227%; i.e., the variation in MSPS was more than twice its mean value. This indicates substantial heterogeneity in switching behavior across drug courses.

In practical terms, this means that for any given maintenance drug therapy initiated with a bioequivalent product, there is approximately a 17% chance that the prescription would be switched to a different manufacturer within a one-year course. As expected, the MSPS rate increased with a greater number of prescription visits (Table 3). However, this increase was not linear; it rose sharply when the number of visits exceeded 12, from approximately 15% for courses with ≤12 visits to 22% for those with >12 visits.

## 3. Sources of MSPS Variation

### 3.1. Drug Characteristics

MSPS rates varied substantially across the 19 drugs included in the analysis (Table 3, Figure 2). Notably, Glimepiride 2 mg shows the highest switching rate (29%), suggesting high market competition or perceived interchangeability. In contrast, Cilnidipine 10 mg has the lowest rate (1%), indicating limited substitution or strong brand preference. Other drugs with higher-than-average MSPS rates included Losartan 50 mg (20%), HCTZ 12.5 mg/Losartan 50 mg (19%)**,** Pioglitazone 15 mg (19%)**,** and HCTZ 12.5 mg/Valsartan 80 mg (18%). In contrast, lower MSPS rates were observed for drugs such as Glibenclamide 5 mg/Metformin 500 mg (3%) and Voglibose 0.2 mg (4%).

The MSPS rate tended to rise for drugs with both greater use and a larger number of bioequivalent sources available in the market, suggesting that market competition and availability play important roles in switching behavior.

### 3.2. Physician Practice Setting and Geographic Region

MSPS also showed marked variation by the physician practice setting (Table 3, Figure 3). The lowest switching rate was observed in tertiary/general hospitals (15%)**,** followed by clinics (17%) and community hospitals (22%)**.** The highest MSPS rate was found in public health center branches (26%)**,** where physicians typically serve as part of mandatory military service. These findings suggest that institutional context and level of prescribing autonomy influence switching rates.

Regional variation was also evident. MSPS was more prevalent in less urbanized regions, where rates ranged from 19% to 20%**,** compared to urban centers such as Seoul (14.3%), Daejeon (14.7%), and Busan (15.0%). This pattern may reflect differences in institutional prescribing practices, access to centralized drug formularies, or local market dynamics.

### 3.3. Patient Characteristics

In contrast to the pronounced variation observed across drug and provider factors, MSPS showed minimal variation by patient characteristics. The switching rate was nearly identical between males and females (17% for both). Age-related differences were also negligible, with the youngest and oldest age groups having a switching rate of 18%, and patients in their 60s slightly lower at 16%. These findings suggest that patient demographics played a limited role in determining whether a switch occurred.

## 4. Discussion

### 4.1. Key Findings and Context

This study found that Korean physicians engaged in multi-source prescription switching (MSPS) in approximately 17% of drug therapy courses for patients with hypertension or diabetes, despite the absence of price differentiation and persistent doubts surrounding generic drug equivalence. Given that all products involved were pharmaceutically equivalent and bioequivalent per regulatory standards, this high switching frequency invites scrutiny regarding its clinical and policy justifications.

Several systemic factors facilitate MSPS in Korea. In contrast to countries like the United States—where generics are typically given INNs—Korean generics are marketed under proprietary tradenames. This enables physicians to specify the proprietary names of individual generic products in their prescriptions, thereby discouraging pharmacies from substituting them with other generics bearing different names. This prescribing environment incentivizes manufacturers to brand their generics in markets like Korea and India, where pharmacy-level substitution is legally restricted or culturally discouraged [21,22]. Although pharmacists in Korea are legally allowed to substitute bioequivalent products, they must notify the prescriber, and due to the dependence of community pharmacies on neighboring clinics, they rarely exercise this right [1,5].

In contrast to Korea’s limited pharmacy-level substitution, the United States consistently achieves high generic substitution rates—often exceeding 90%—through a combination of regulatory, financial, and institutional levers. These include robust market competition, lower development and approval costs for generics, and tiered reimbursement incentives by insurers that favor generics over brand-name drugs. Additionally, state-level laws in the U.S. typically mandate automatic substitution at the pharmacy unless explicitly disallowed by the prescriber. These substitution practices are further reinforced by pharmacy benefit managers (PBMs), who manage formularies and negotiate pricing to prioritize cost-effective dispensing. This system-wide alignment stands in contrast to Korea’s uniform pricing model and more permissive, discretionary substitution policy, helping explain the persistently low generic substitution rates despite legal allowance. These cross-national differences underscore the importance of aligning policy incentives, provider behavior, and formulary mechanisms to improve substitution outcomes.

While this infrastructure facilitates MSPS, its clinical or economic justification remains weak. First, there is widespread skepticism among Korean physicians regarding the interchangeability of generics [4]. Many perceive generic drugs as inferior “copycats,” raising concerns about their quality and reliability [23,24,25,26]. Second, physicians face cognitive and logistical burdens when prescribing generics: they must navigate a crowded marketplace of similarly formulated products, evaluate their perceived interchangeability, and accurately recall proprietary names. Third, under Korea’s uniform drug pricing system, there is no financial incentive for switching, as price differences between originator brands and generic counterparts are negligible. Collectively, these factors weaken both the clinical rationale and the economic justification for MSPS.

While MSPS offers no clear advantage to patients or the healthcare system, it may occur when patients see different physicians. However, MSPS resulting from physician changes undermines the principles of continuity of care and patient-centeredness. For therapies that are effective, switching to a different manufacturer may disrupt treatment stability—particularly in a context where physicians themselves question the clinical equivalence of alternatives. For therapies that are ineffective, a therapeutic class switch—not a manufacturer switch—would be more appropriate. Moreover, because MSPS involves changes in tradenames, it may lead to patient confusion or anxiety. Prior research has shown that even changes in pill appearance can reduce adherence; thus, changes in drug names may further exacerbate this issue [27].

Substantial variation in MSPS rates across drug types, physician practice settings, and regions underscores the discretionary nature of the practice. Switching rates ranged from as low as 1% for cilnidipine to as high as 29% for glimepiride, suggesting differences in perceived interchangeability, competitive market dynamics, or promotional influence. MSPS was notably more prevalent in less integrated practice environments and government-run health centers, where individual prescribers exercise greater autonomy. By contrast, integrated health systems with centralized oversight—such as those employing Pharmacy and Therapeutics (P&T) committees—demonstrated lower switching rates, likely due to standardized formulary management and institutional prescribing guidelines [28].

These findings are consistent with a broader body of research on physician-driven variation in clinical practice. Prior studies have demonstrated that individual practice styles significantly shape clinical behaviors—from primary care utilization [29] to cesarean section rates [30] and preoperative testing decisions [31]—particularly in contexts of clinical uncertainty [32]. In Korea, where skepticism about the equivalence of generic drugs remains pervasive, MSPS may similarly reflect physician-specific prescribing patterns rather than adherence to evidence-based protocols. Future research should investigate whether such switching behavior is primarily influenced by clinical uncertainty, promotional activities, or institutional shortcomings in governance.

In addition to physician discretion and market competition, other system-level factors may also contribute to MSPS. Institutional purchasing policies, such as volume-based procurement agreements or supply contracts, can limit product selection within multi-source drug markets. Similarly, periodic drug shortages may force physicians to prescribe alternative but bioequivalent products. Moreover, formulary changes driven by administrative or financial considerations may intermittently affect the availability of certain generics or originator products. These structural influences warrant further investigation to fully understand the determinants of MSPS in Korea’s pharmaceutical system.”

From a behavioral perspective, the patterns of MSPS observed in this study can be interpreted through the Theory of Planned Behavior (TPB)**,** which posits that individual actions are shaped by three main constructs: attitudes, subjective norms, and perceived behavioral control [33]. In the context of physician prescribing, attitudes may reflect skepticism toward the clinical equivalence of generics, despite their regulatory approval. Subjective norms could stem from the influence of peers, institutional culture, or pharmaceutical marketing, which subtly shape prescribing behaviors. Lastly, perceived behavioral control may be affected by institutional formularies, supply constraints, or ambiguity in national policy regarding generic substitution. Together, these factors may explain the discretionary and variable nature of MSPS, reinforcing the need for policies that not only address economic incentives but also behavioral drivers [34,35].

### 4.2. Policy Implications

The findings suggest that Korea’s current pharmacy benefit structure—particularly its uniform pricing model and permissive substitution policy—does not sufficiently align prescriber behavior with national goals of cost-efficiency and consistent prescribing. In the absence of price differentials and amid persistent doubts about the equivalence of generics, physicians have little incentive to optimize drug selection based on clinical or economic value.

International experience offers useful direction: the UK mandates international nonproprietary name (INN)-based prescribing, Sweden enforces centralized reimbursement and mandatory substitution, and Australia promotes rational prescribing through structured education and competency frameworks. These approaches may help reduce unnecessary switching and improve the consistency of multi-source drug use.

To adapt such strategies to Korea’s mixed practice environment, differentiated interventions may be needed: In integrated care settings (e.g., tertiary and general hospitals), policymakers could consider leveraging institutional mechanisms such as Pharmacy and Therapeutics (P&T) committees and standardized formularies to promote INN-based prescribing, disseminate evidence on bioequivalence, and implement centralized prescribing protocols to reduce discretionary switching. In independent practice settings (e.g., private clinics, public health centers), strategies may include expanded provider education on generic quality, feedback on prescribing patterns, and pilot testing of accountability tools such as audit reports or financial disclosure mechanisms. Reinforcing compliance with anti-kickback laws and exploring public reporting initiatives could further discourage brand-driven prescribing.

Overall, these findings underscore the need to explore targeted policy levers that promote clinically sound and economically rational prescribing practices. Future research and pilot programs could assess the effectiveness of these interventions in curbing avoidable MSPS within Korea’s unique uniform-pricing environment.

### 4.3. Key Policy Considerations:

Promote INN-based prescribing in institutional settings through formulary alignment and clinical governance (e.g., P&T committees).Enhance physician education on bioequivalence and prescribing standards in independent practice settings.Explore pilot programs for prescribing audits and transparency mechanisms (e.g., reporting of switching patterns or financial disclosures).Consider centralized procurement or formulary standardization strategies, drawing on international models (e.g., UK, Sweden, Australia).

### 4.4. Limitations

Several limitations should be acknowledged. First, the study did not differentiate whether the index drug was an originator or a generic. While MSPS may differ by initial product type, our primary interest was in switching behavior regardless of originator status. Second, the analysis did not account for continuity of care; some observed switching may have resulted from changes in prescribers rather than within-provider variation. Third, the use of claims data limited insight into the clinical rationale behind prescribing decisions and provided no information on physicians’ exposure to pharmaceutical promotion. Fourth, although therapy courses were constructed based on prescription records, actual medication adherence or use could not be verified. Fifth, the analysis was limited to older adults with hypertension or diabetes, which may reduce generalizability to younger populations or other therapeutic classes. Sixth, this study’s findings may not be generalizable beyond Korea’s policy environment. Additionally, we could not account for time-varying influences such as formulary updates, drug shortages, and supply chain disruptions—nor access physician-level variables (e.g., tenure, prescribing style) or patient socioeconomic factors—that may have influenced switching behavior.

Finally, MSPS was defined based on changes in product codes within bioequivalent drug classes, assuming these changes reflected intentional source switching by prescribers. However, some switches may have been driven by non-clinical factors—such as formulary adjustments, drug shortages, or institutional purchasing policies—that still required physician sign-off but may not have reflected individualized clinical judgment.

## 5. Conclusions

This study examined the extent and variation in multi-source prescription switching (MSPS) among older adults with hypertension or diabetes in Korea’s uniform-pricing pharmaceutical system. Despite limited generic substitution at the pharmacy level and the absence of meaningful price differentials among equivalent products, physicians frequently initiated MSPS during maintenance therapy. The findings suggest that physician prescribing behavior—rather than patient characteristics or clinical necessity—is the primary driver of MSPS, influenced by the availability of branded generics, varying levels of institutional oversight, and competitive market dynamics.

Given the absence of clear clinical or economic justification, and given that such switching could potentially influence patient trust and adherence, MSPS may point to underlying inefficiencies in Korea’s pharmaceutical benefit design that warrant further exploration. Addressing this inefficiency will require multifaceted policy reforms, including promotion of INN-based prescribing, reduction in proprietary naming of generics, and stronger oversight of physician–industry relationships. These measures will be essential to align prescribing practices with principles of patient-centered care and cost-effective health system management.

## Figures and Tables

**Figure 1 healthcare-13-02339-f001:**
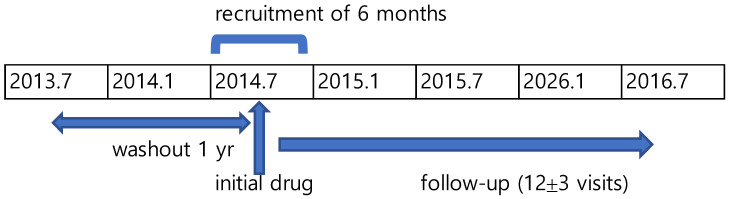
Study Observation Period.

**Figure 2 healthcare-13-02339-f002:**
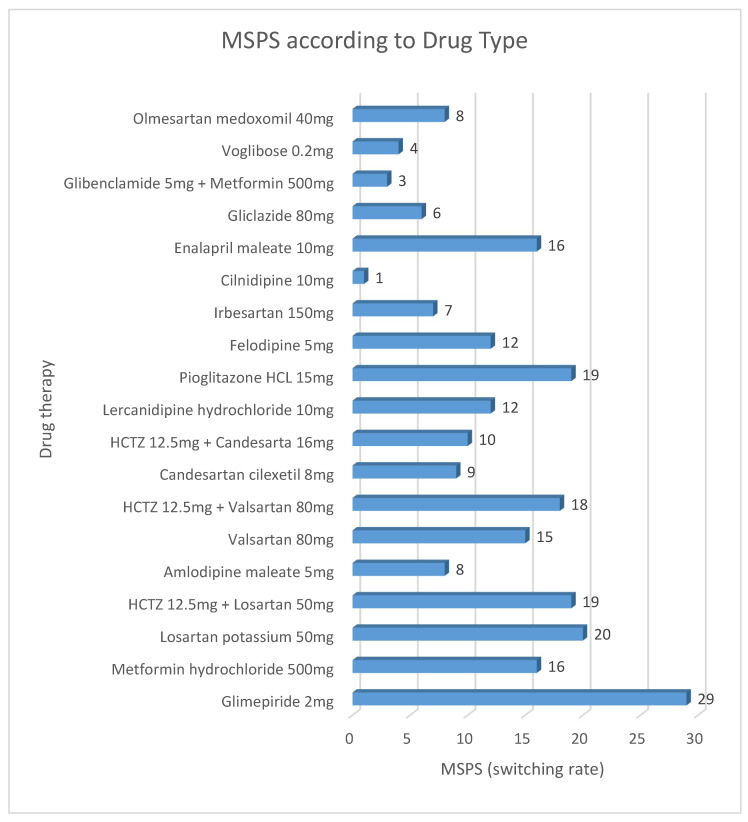
MSPS according to Drug Type.

**Figure 3 healthcare-13-02339-f003:**
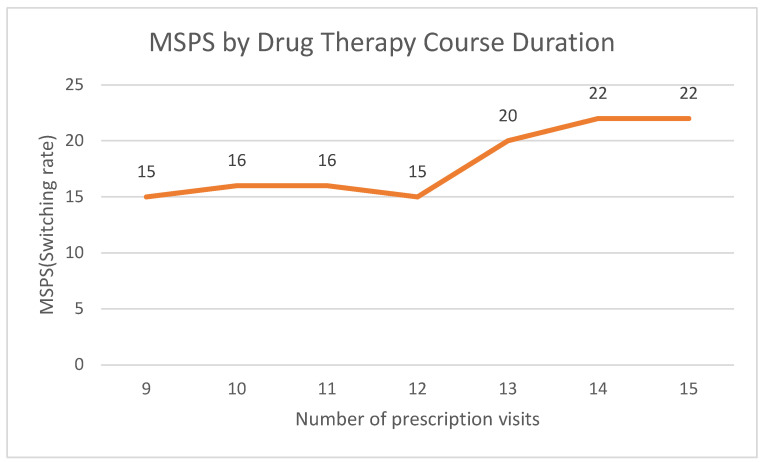
MSPS by Drug Therapy Course Duration (Number of Prescription Visits).

**Table 1 healthcare-13-02339-t001:** Difference in generic drugs policies and practices between Korea and the US.

Issues	US	Korea
Substitutability Concern	Original vs. Generic	One Generic vs. Another
Names of Generic Drugs	INN (International Non-proprietary Name)	Proprietary Name
Prescribing	Mostly the Original’s Brand Name	The Original’s and Branded Generics’ Proprietary Names
Reputation of Generic Drug Manufacturers	Mostly None	Quite reputable for the top 10 Korean manufacturers
Barriers to Substitution	DAW (Dispense as written)	DAW and Requiring the pharmacy to notify the physician of the substitution.
Generic substitution at the pharmacy	Quite Often	Rare
Price Differential between Originators and Generics	Substantial	Negligible
Terminology for compensating physicians for prescribing certain drugs	Kickback	Rebate ^‡^
Physician Perception on Bioequivalence	Positive Side	Negative Side
Pharmacy Practices	Mainly Large Corporate Practice	Small-Scale Independent Practice

^‡^: Historically, Korean physicians dispensed medications directly at their clinics until a legal mandate requiring the separation of prescribing and dispensing practices was introduced. Under the previous system, physicians commonly received rebates from drug manufacturers in exchange for purchasing and dispensing specific products. Although this direct purchasing arrangement is no longer relevant for drugs dispensed through pharmacies, the term “rebate” continues to be used—albeit inaccurately—to describe inducements offered to physicians for prescribing certain drugs in the post-reform era.

**Table 2 healthcare-13-02339-t002:** Basic characteristics of drug therapies.

Variables		Freq. (N)	Percent (%)
Gender	Male	646,191	48.76
	Female	679,143	51.24
Age	<50	115,146	8.69
	50s	327,368	24.70
	60s	405,289	30.58
	70s	348,742	26.31
	≥80	128,789	9.72
Insurance Type	NHIS	1,244,245	93.88
	Medical Aid	78,167	5.90
	Veteran	2922	0.22
Number of Sources *	≥75	749,292	56.54
	50–75	316,603	23.89
	25–50	171,335	12.93
	<25	88,104	6.65
Total		1,325,334	100.00

* 75~109: Amlodipine maleate 5 mg, Glimepiride 2 mg, Glibenclamide 5 mg + Metformin 500 mg, Losartan potassium 50 mg, Pioglitazone HCl 15 mg, Metformin hydrochloride 500 mg; 50~65: Enalapril maleate 10 mg, Olmesartan medoxomil 40 mg, HCTZ 12.5 mg + Losartan 50 mg, Valsartan 80 mg, HCTZ 12.5 mg + Valsartan 80 mg; 23~34: Felodipine 5 mg, Lercanidipine HCl 10 mg, HCTZ 12.5 mg + Candesarta 16 mg, Irbesartan 150 mg; 6~17: Voglibose 0.2 mg, Gliclazide 80 mg, Candesartan cilexetil 8 mg, Cilnidipine 10 mg.

**Table 3 healthcare-13-02339-t003:** MSPS by drug therapy, number of prescription visits and type of institution.

Variables	% Drug Therapies		MSPS (Switching Rate per 100)		
	Freq. (N)	Percent (%)	Mean	S.D.	C.V.
Selected Formulations of BE Drug Therapy					
Glimepiride 2 mg	213,341	16.1	29	45	1.55
Metformin hydrochloride 500 mg	184,018	13.88	16	37	2.31
Losartan potassium 50 mg	176,068	13.28	20	40	2.00
HCTZ 12.5 mg + Losartan 50 mg	156,432	11.8	19	39	2.05
Amlodipine maleate 5 mg	123,940	9.35	8	28	3.50
Valsartan 80 mg	73,884	5.57	15	36	2.40
HCTZ 12.5 mg + Valsartan 80 mg	67,028	5.06	18	38	2.11
Candesartan cilexetil 8 mg	55,031	4.15	9	29	3.22
HCTZ 12.5 mg + Candesartan 16 mg	50,110	3.78	10	30	3.00
Lercanidipine hydrochloride 10 mg	46,753	3.53	12	33	2.75
Pioglitazone HCL 15 mg	45,600	3.44	19	39	2.05
Felodipine 5 mg	38,785	2.93	12	33	2.75
Irbesartan 150 mg	35,687	2.69	7	26	3.71
Cilnidipine 10 mg	16,424	1.24	1	10	10.00
Enalapril maleate 10 mg	15,133	1.14	16	37	2.31
Gliclazide 80 mg	11,480	0.87	6	23	3.83
Glibenclamide 5 mg + Metformin 500 mg	6325	0.48	3	17	5.67
Voglibose 0.2 mg	5169	0.39	4	18	4.50
Olmesartan medoxomil 40 mg	4126	0.31	8	28	3.50
Number of Prescription Visits					
9	202,936	15.31	15	36	2.40
10	191,739	14.47	16	37	2.31
11	229,847	17.34	16	37	2.31
12	318,685	24.05	15	36	2.40
13	165,309	12.47	20	40	2.00
14	114,699	8.65	22	41	1.86
15	102,119	7.71	22	42	1.91
Type of Physician Practices					
Tertiary Hospital	58,986	4.45	15	36	2.40
General Hospital	169,065	12.76	16	36	2.25
Hospital	78,657	5.93	22	42	1.91
Eldercare Hospital	9986	0.75	22	42	1.91
Clinic	907,988	68.51	17	37	2.18
Public Health Center	76,626	5.78	17	38	2.24
Public Health Center Branch *	20,236	1.53	26	44	1.69
Public Healthcare Facility **	3790	0.29	23	42	1.83

*: Physicians were serving as part of military duty. **: Nurses were serving to eliminate the rural counties with no physician service.

## Data Availability

The data that support the findings of this study are available at HIRA, but restrictions apply to the availability of these data, which were used under license for the current study. Although the data are not publicly available, they can be licensed again for analysis with permission from HIRA.

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
