# Peer review of "Patterns of Prescription Switching in a Uniform-Pricing System for Multi-Source Drugs: A Retrospective Population-Based Cohort Study"

_healthcare, 2025, doi:10.3390/healthcare13182339_

Round 1

Reviewer 1 Report

Comments and Suggestions for Authors

Thank you for the opportunity to review this timely manuscript. Your work provides valuable insights into physician-driven prescription switching in Korea’s uniform-pricing pharmaceutical system. The study is well-conceived and contributes meaningfully to the literature on generic drug utilization and policy design. With some refinements this manuscript has strong potential to inform future research and health policy development. 

  1. Abstract - Consider revising the Background section to include a more explicit discussion of the observed variation in MSPS across provider settings and drug markets. Specifically, switching behavior varies significantly by institutional context and drug type, but not by patient characteristics, highlighting the discretionary nature of prescribing and suggesting a need for targeted pharmacy benefit policies.
  2. Introduction - Add references to support the claims made regarding pharmacists’ limited authority and behavior in generic substitution. Specifically, the statements about pharmacists rarely exercising substitution rights due to concerns about relationships with physicians, and the extremely low substitution rate at the pharmacy level (0.2%), would benefit from further citation. 
  3. Introduction - Add a reference for the statement regarding the structure and behavior of pharmacies in Korea. Specifically, the claim that most pharmacies are small, independently owned businesses that rely heavily on prescriptions from nearby clinics, and therefore refrain from exercising substitution rights to preserve relationships with prescribers, would benefit from citation. 
  4. Methods - The methods do not mention any attempt to control for potential confounders (e.g., prescriber characteristics, institutional policies). Please acknowledge this limitation in the Methods or Discussion section to strengthen the manuscript.
  5. Methods - You state that a drug therapy course is defined as 12 ± 3 prescription visits. It would be helpful to explain why this range was chosen and how it reflects real-world prescribing patterns.
  6. Methods - The exclusion of drug markets with fewer than four bioequivalent competitors is mentioned. Please explain the justification for this.
  7. Figures 2 and 3 - It will be helpful if you comment on the broader patterns shown in the figures within the text. For example, 'Figure 2 illustrates the variation in MSPS rates across different drug types. Notably, Glimepiride 2 mg shows the highest switching rate (29%), suggesting high market competition or perceived interchangeability. In contrast, Cilnidipine 10 mg has the lowest rate (1%), indicating limited substitution or strong brand preference. These differences may reflect variations in physician confidence, promotional activity, or formulary availability across therapeutic classes.'
  8. Discussion - Consider softening the tone and scope of the Policy Implications section. While the recommendations are thoughtful and well-grounded in the study’s findings, some of the proposed reforms may come across as overly prescriptive or ambitious given the observational nature of the study. You might consider framing these suggestions as potential areas for further exploration or pilot testing.
  9. Discussion - To strengthen the Policy Implications section, consider including examples from other countries that have successfully reduced MSPS through specific interventions (e.g., formulary standardization, INN-based prescribing mandates, or centralized procurement strategies).
  10. Discussion - Expand the discussion of factors contributing to MSPS beyond physician discretion and market dynamics. While these are important drivers, other influences, such as institutional purchasing policies, drug shortages, and formulary changes, may also play a role in switching behavior. 
  11. Discussion - To enhance the policy discussion, consider incorporating additional context on why generic substitution rates are higher in the U.S. Specifically, you could note that U.S. substitution is supported by lower R&D costs for generics, strong market competition, and robust incentives from insurance companies and state-level policies. Including this either in Table 1 or the Discussion could highlight the systemic differences and help clarify why Korea’s substitution rates remain low despite legal permissibility.
  12. Limitations - Additional limitations that can be noted include lack of generalizability beyond Korea, changes in prescribers or institutional policies (e.g., formulary updates or drug shortages) that might have influenced switching, as well as lack of clinical data or physician-level variables that could help explain prescribing decisions.

Author Response

Thank you for the opportunity to review this timely manuscript. Your work provides valuable insights into physician-driven prescription switching in Korea’s uniform-pricing pharmaceutical system. The study is well-conceived and contributes meaningfully to the literature on generic drug utilization and policy design. With some refinements this manuscript has strong potential to inform future research and health policy development. 

  1. Abstract - Consider revising the Background section to include a more explicit discussion of the observed variation in MSPS across provider settings and drug markets. Specifically, switching behavior varies significantly by institutional context and drug type, but not by patient characteristics, highlighting the discretionary nature of prescribing and suggesting a need for targeted pharmacy benefit policies.

Response

We thank the reviewer for this insightful comment. We revised the Background section of the Abstract to more clearly highlight the observed variation in MSPS across provider settings and drug markets. By linking MSPS to discretionary prescribing behavior, we sought to emphasize its relevance for policy, particularly the need for targeted pharmacy benefit strategies that strengthen substitutability and promote effective competition among multi-source drugs.

Revised Background in Abstract (with changes in bold):

Generic drugs account for approximately 40% of the Korean prescription drug market, despite limited generic substitution at the point of dispensing. This suggests that switching between originator and generic drugs often occurs at the point of prescription. Physicians, in fact, have opposed pharmacy-level substitution due to concerns about the clinical equivalence of generics, despite the regulatory confirmation of their bioequivalence. Importantly, multi-source prescription switching (MSPS) may reflect discretionary prescribing behavior, underscoring the need for targeted benefit policies to enhance substitutability and promote effective competition among multi-source drugs. This study aimed to quantify the extent of physician-initiated MSPS among adults with hypertension or diabetes and to identify factors associated with these switching behaviors

  1. Introduction - Add references to support the claims made regarding pharmacists’ limited authority and behavior in generic substitution. Specifically, the statements about pharmacists rarely exercising substitution rights due to concerns about relationships with physicians, and the extremely low substitution rate at the pharmacy level (0.2%), would benefit from further citation. 

Response

We appreciate the reviewer’s comment. In response, we added citations to support our claims regarding pharmacists’ limited authority and behavior in exercising substitution rights. Specifically, we cited data indicating the extremely low substitution rate of approximately 0.2% over four year

  1. Introduction - Add a reference for the statement regarding the structure and behavior of pharmacies in Korea. Specifically, the claim that most pharmacies are small, independently owned businesses that rely heavily on prescriptions from nearby clinics, and therefore refrain from exercising substitution rights to preserve relationships with prescribers, would benefit from citation. 

Response

We appreciate the reviewer’s suggestion. In response, we added citations to support the characterization of Korean pharmacies as predominantly small, independently owned businesses that depend heavily on prescription traffic from nearby clinics.

  1. Methods - The methods do not mention any attempt to control for potential confounders (e.g., prescriber characteristics, institutional policies). Please acknowledge this limitation in the Methods or Discussion section to strengthen the manuscript.

Response

We thank the reviewer for this important observation. In response to the reviewer’s suggestion, we have added a statement in the Discussion section acknowledging this limitation explicitly.

Revised limitation

Additionally, we could not account for time-varying influences such as formulary up-dates, drug shortages, and supply chain disruptions—nor access physician-level variables (e.g., tenure, prescribing style) or patient socioeconomic factors—that may have influenced switching behavior

  1. Methods - You state that a drug therapy course is defined as 12 ± 3 prescription visits. It would be helpful to explain why this range was chosen and how it reflects real-world prescribing patterns.

Response

We appreciate the reviewer’s thoughtful suggestion. In response, we have supplemented the Methods section with statistical information on the average days of supply (33.5 days) for diabetic patients in Korea. This addition helps justify our operational definition of a one-year drug therapy course (12 ±â€¯3 visits). We have also added a supporting citation accordingly.

Revised text

A drug therapy course of 12 ±â€¯3 prescription visits likely approximates a one-year treatment period, given the average of 33.5 days per prescription for diabetic patients under the Korean health insurance system (Kim, 2017).

  1. Methods - The exclusion of drug markets with fewer than four bioequivalent competitors is mentioned. Please explain the justification for this.

Response

The exclusion of drug markets with fewer than four bioequivalent competitors was required because the NHIS de-identified individual product codes in such cases to safeguard the commercial confidentiality of manufacturers, which could otherwise be compromised.

Revised text

Only drugs with at least four bioequivalent competitors were included, due to data availability restrictions from the Health Insurance Review and Assessment Service (HIRA), which withholds manufacturer-level prescription data for drug markets with fewer than four competitors to safeguard confidentiality.

  1. Figures 2 and 3 - It will be helpful if you comment on the broader patterns shown in the figures within the text. For example, 'Figure 2 illustrates the variation in MSPS rates across different drug types. Notably, Glimepiride 2 mg shows the highest switching rate (29%), suggesting high market competition or perceived interchangeability. In contrast, Cilnidipine 10 mg has the lowest rate (1%), indicating limited substitution or strong brand preference. These differences may reflect variations in physician confidence, promotional activity, or formulary availability across therapeutic classes.'

Response

We appreciate the reviewer’s helpful suggestion and have gladly replaced our original sentence with the recommended wording for Figure 2. However, we chose not to include the final clause suggesting variations in physician confidence, promotional activity, or formulary availability across therapeutic classes. We sought to avoid adding attributive content to the Results section, as we believe such explanations are more appropriate for the Discussion. That said, we retained the reviewer’s phrasing “indicating limited substitution or strong brand preference” as a minimally interpretive descriptor that remains within the descriptive scope of the Results.

Regarding Figure 3, we confirm that it is already clearly described in the manuscript. We also intentionally avoided a redundant introductory phrase such as “Figure 3 illustrates…” because the relationship between the narrative and the figure is already explicit in the text.

We hope this clarifies our editorial choices and are grateful for the thoughtful feedback.

Revised Text:

“Notably, glimepiride 2 mg shows the highest switching rate (29%), suggesting high market competition or a high degree of perceived interchangeability. In contrast, cilnidipine 10 mg has the lowest rate (1%), indicating limited substitution or strong brand preference.”

  1. Discussion - Consider softening the tone and scope of the Policy Implications section. While the recommendations are thoughtful and well-grounded in the study’s findings, some of the proposed reforms may come across as overly prescriptive or ambitious given the observational nature of the study. You might consider framing these suggestions as potential areas for further exploration or pilot testing.

Response

We thank the reviewer for this insightful and constructive feedback. We agree that, given the observational and descriptive nature of our study, policy recommendations should be presented with appropriate caution to avoid implying causal conclusions or overstating the generalizability of our findings.

In response, we have revised the Policy Implications section in the Discussion to adopt a more exploratory tone.

Revised text

The findings suggest that Korea’s current pharmacy benefit structure—particularly its uniform pricing model and permissive substitution policy—does not sufficiently align prescriber behavior with national goals of cost-efficiency and consistent prescribing. In the absence of price differentials and amid persistent doubts about the equivalence of generics, physicians have little incentive to optimize drug selection based on clinical or economic value.

International experience offers useful direction: the UK mandates international nonproprietary name (INN)-based prescribing, Sweden enforces centralized reimbursement and mandatory substitution, and Australia promotes rational prescribing through structured education and competency frameworks. These approaches may help reduce unnecessary switching and improve the consistency of multi-source drug use.

To adapt such strategies to Korea’s mixed practice environment, differentiated interventions may be needed: In integrated care settings (e.g., tertiary and general hospitals), policymakers could consider leveraging institutional mechanisms such as Pharmacy and Therapeutics (P&T) committees and standardized formularies to promote INN-based prescribing, disseminate evidence on bioequivalence, and implement centralized prescribing protocols to reduce discretionary switching. In independent practice settings (e.g., private clinics, public health centers), strategies may include expanded provider education on generic quality, feedback on prescribing patterns, and pilot testing of accountability tools such as audit reports or financial disclosure mechanisms. Reinforcing compliance with anti-kickback laws and exploring public reporting initiatives could further discourage brand-driven prescribing.

Overall, these findings underscore the need to explore targeted policy levers that promote clinically sound and economically rational prescribing practices. Future research and pilot programs could assess the effectiveness of these interventions in curbing avoidable MSPS within Korea’s unique uniform-pricing environment.

  1. Discussion - To strengthen the Policy Implications section, consider including examples from other countries that have successfully reduced MSPS through specific interventions (e.g., formulary standardization, INN-based prescribing mandates, or centralized procurement strategies).

Response

We thank the reviewer for this valuable suggestion. We agree that including international examples can provide useful context and strengthen the policy relevance of our findings. In response, we have revised the Policy Implications section to incorporate selected examples from other countries that have implemented effective interventions to reduce unnecessary multi-source prescription switching (MSPS) and promote rational prescribing.

Revised text

International experience offers useful direction: the UK mandates international nonproprietary name (INN)-based prescribing, Sweden enforces centralized reimbursement and mandatory substitution, and Australia promotes rational prescribing through structured education and competency frameworks. These approaches may help reduce unnecessary switching and improve the consistency of multi-source drug use.

  1. Discussion - Expand the discussion of factors contributing to MSPS beyond physician discretion and market dynamics. While these are important drivers, other influences, such as institutional purchasing policies, drug shortages, and formulary changes, may also play a role in switching behavior. 

Response

We thank the reviewer for highlighting this important point. We agree that physician discretion and market dynamics, while central to our analysis, do not fully account for all the potential drivers of multi-source prescription switching (MSPS). In response, we have expanded the Discussion section to acknowledge additional factors that may influence switching behavior.

Specifically, we now note that institutional purchasing policies, such as bulk procurement or supply contracts negotiated at the hospital or health system level, may constrain the choice of specific bioequivalent products regardless of physician preference. Drug shortages, whether temporary or systemic, can also necessitate switching across multi-source products. Finally, formulary changes, either due to administrative review or pricing negotiations, may affect the availability of certain products over time, particularly in settings where formularies are centrally managed.

We have added the following text to the Discussion section. We appreciate the reviewer’s suggestion, which has allowed us to present a more comprehensive interpretation of the observed switching behavior.

Revised text

“In addition to physician discretion and market competition, other system-level factors may also contribute to MSPS. Institutional purchasing policies, such as volume-based procurement agreements or supply contracts, can limit product selection within multi-source drug markets. Similarly, periodic drug shortages may force physicians to prescribe alternative but bioequivalent products. Moreover, formulary changes driven by administrative or financial considerations may intermittently affect the availability of certain generics or originator products. These structural influences warrant further investigation to fully understand the determinants of MSPS in Korea’s pharmaceutical system.”

  1. Discussion - To enhance the policy discussion, consider incorporating additional context on why generic substitution rates are higher in the U.S. Specifically, you could note that U.S. substitution is supported by lower R&D costs for generics, strong market competition, and robust incentives from insurance companies and state-level policies. Including this either in Table 1 or the Discussion could highlight the systemic differences and help clarify why Korea’s substitution rates remain low despite legal permissibility.

Response

We thank the reviewer for this helpful suggestion. We agree that contrasting Korea’s low generic substitution rate with the structural drivers of higher substitution in the U.S. provides important context for interpreting our findings.

In response, we have added a brief comparative paragraph to the Discussion section to highlight how several U.S. system-level factors—such as lower R&D burdens for generics, stronger market competition, formulary-tiered pricing, and insurance-driven substitution incentives—contribute to consistently high rates of generic use. We have also noted that state-level substitution mandates, combined with pharmacy benefit manager (PBM) enforcement, further promote substitution at the dispensing level.

Revised Text

In contrast to Korea’s limited pharmacy-level substitution, the United States consistently achieves high generic substitution rates—often exceeding 90%—through a combination of regulatory, financial, and institutional levers. These include robust market competition, lower development and approval costs for generics, and tiered reimbursement incentives by insurers that favor generics over brand-name drugs. Additionally, state-level laws in the U.S. typically mandate automatic substitution at the pharmacy unless explicitly disallowed by the prescriber. These substitution practices are further reinforced by pharmacy benefit managers (PBMs), who manage formularies and negotiate pricing to prioritize cost-effective dispensing. This system-wide alignment stands in contrast to Korea’s uniform pricing model and more permissive, discretionary substitution policy, helping explain the persistently low generic substitution rates despite legal allowance. These cross-national differences underscore the importance of aligning policy incentives, provider behavior, and formulary mechanisms to improve substitution outcomes.

  1. Limitations - Additional limitations that can be noted include lack of generalizability beyond Korea, changes in prescribers or institutional policies (e.g., formulary updates or drug shortages) that might have influenced switching, as well as lack of clinical data or physician-level variables that could help explain prescribing decisions.

Response

We agree with the reviewer and have added text to the Limitations section noting that our findings may not generalize beyond Korea, and that we could not account for time-varying factors (e.g., formulary changes, drug shortages) or access physician-level or clinical data that might explain switching behavior more precisely.

Revised text:

Sixth, this study’s findings may not be generalizable beyond Korea’s policy environment. Additionally, we could not account for time-varying influences such as formulary updates, drug shortages, and supply chain disruptions—nor access physician-level variables (e.g., tenure, prescribing style) or patient socioeconomic factors—that may have influenced switching behavior

Reviewer 2 Report

Comments and Suggestions for Authors

The manuscript that is presented to me here shows an important and underexplored topic, providing valuable insights into multi-source prescription switching (MSPS) patterns in Korea’s uniform-pricing system. I can see that the intro is well-structured, clearly showing me (or other readers) the policy context and research gap, and is supported by relevant and up-to-date references. The way the authors designed this study is very good for the stated objectives. The methods shown here are shown in detail; I have a feeling that I could easily reproduce this study, which is very important. 

I have one major issue regarding this study - I simply can not find anything to make it better or find any flaws that I could recommedn to change. This is awkward situation for reviewers because we like to think that we simply know a lot more than the authors themselves, which is often not true. So I would like to congratulate the authors on this very exquisit study. 

Results are clearly presented, with appropriate use of tables and figures to illustrate variations in MSPS across drugs, settings, and regions. The discussion effectively interprets the findings within both the Korean and international contexts, linking them to policy implications.

The quality of English is good (there are some long sentences, but no majo issues). 

The only things I would suggest:

- a brief paragraph summarizing the key policy recommendations in bullet form could make the implications section more actionable for policymakers

- the limitations are clearly acknowledged, but the potential influence of unmeasured variables such as physician tenure, patient socioeconomic status, or supply chain disruptions could be briefly noted as avenues for future research

Whn you take everything into consideration this is a well-prepared and relevant study that makes a meaningful contribution to understanding prescription behaviors in multi-source drug markets.

Author Response

Reviewer2 Comment:

The manuscript that is presented to me here shows an important and underexplored topic, providing valuable insights into multi-source prescription switching (MSPS) patterns in Korea’s uniform-pricing system. I can see that the intro is well-structured, clearly showing me (or other readers) the policy context and research gap, and is supported by relevant and up-to-date references. The way the authors designed this study is very good for the stated objectives. The methods shown here are shown in detail; I have a feeling that I could easily reproduce this study, which is very important. 

I have one major issue regarding this study - I simply can not find anything to make it better or find any flaws that I could recommedn to change. This is awkward situation for reviewers because we like to think that we simply know a lot more than the authors themselves, which is often not true. So I would like to congratulate the authors on this very exquisit study. 

Results are clearly presented, with appropriate use of tables and figures to illustrate variations in MSPS across drugs, settings, and regions. The discussion effectively interprets the findings within both the Korean and international contexts, linking them to policy implications.

The quality of English is good (there are some long sentences, but no majo issues). 

Response

We sincerely thank the reviewer for their kind words and generous feedback. We are truly honored by your appreciation and encouraged by your support of our work.

Comment:

The only things I would suggest:

- a brief paragraph summarizing the key policy recommendations in bullet form could make the implications section more actionable for policymakers

Response

Certainly! Here's a concise paragraph summarizing the key policy recommendations from your revised policy implications section, in bullet-point format, to make them more actionable for policymakers, as the reviewer requested:

Revised text

Key Policy Considerations:

  • Promote INN-based prescribing in institutional settings through formulary alignment and clinical governance (e.g., P&T committees).
  • Enhance physician education on bioequivalence and prescribing standards in independent practice settings.
  • Explore pilot programs for prescribing audits and transparency mechanisms (e.g., reporting of switching patterns or financial disclosures).
  • Consider centralized procurement or formulary standardization strategies, drawing on international models (e.g., UK, Sweden, Australia).

Comment: the limitations are clearly acknowledged, but the potential influence of unmeasured variables such as physician tenure, patient socioeconomic status, or supply chain disruptions could be briefly noted as avenues for future research

Response

We thank the reviewer for this valuable suggestion. In response, we have added a sentence to the Limitations section noting that unmeasured factors such as physician tenure, patient socioeconomic status, and supply chain disruptions may also influence switching behavior and represent important directions for future research.

Revised text

Additionally, we could not account for time-varying influences such as formulary updates, drug shortages, and supply chain disruptions—nor access physician-level variables (e.g., tenure, prescribing style) or patient socioeconomic factors—that may have influenced switching behavior

Comment

Whn you take everything into consideration this is a well-prepared and relevant study that makes a meaningful contribution to understanding prescription behaviors in multi-source drug markets.

Response

We greatly appreciate the reviewer’s thoughtful and encouraging feedback. It is very rewarding to know that the study is seen as a meaningful contribution to the understanding of prescription behaviors in multi-source drug markets. Thank you for your generous assessment

Reviewer 3 Report

Comments and Suggestions for Authors

Introduction

The author should explain the reason for choosing a hypertension or diabetes diagnosis.

Is there any national policy of switching between originator and generic drugs in Korea? Is there any national policy for physicians to only write generic drugs? Is there any policy to guarantee the quality of generic drugs? Is there any ‘me too’ drugs in Korea? Is there any policy for the maximum price of drugs in the market?

The author should add any theory to explain the phenomenon, such as the theory of planned behavior (TPB). This study’s objective is physician behavior. Many factors affect drug prices, not only physician behavior, but also payer, availability (logistic), and regulation (Lee, 2020; Zarei, 2023).

Reference:

Lee KS, Kassab YW, Taha NA, Zainal ZA. Factors Impacting Pharmaceutical Prices and Affordability: Narrative Review. Pharmacy (Basel). 2020 Dec 23;9(1):1. doi: 10.3390/pharmacy9010001. PMID: 33374493; PMCID: PMC7838942.

Zarei E, Ghaffari A, Nikoobar A, Bastami S, Hamdghaddari H. Interaction between physicians and the pharmaceutical industry: A scoping review for developing a policy brief. Front Public Health. 2023 Jan 12;10:1072708. doi: 10.3389/fpubh.2022.1072708. PMID: 36711334; PMCID: PMC9879663.

Methods

Did the author do factor analysis? What is the dominant factor affects the switching drug behavior?

Results

In Table 1: What does the author mean by ‘Negligible’ for ‘Price Differential between Brands & Generics’?

In Table 2: What is ‘N. of Sources’?

This study evaluates switching between originator and generic drugs. Is it only for the same or for different active substances, too?

Discussion

The author should discuss Table 1 comprehensively. Information in Table 1 implies that no drug quality issue because the generic drug is produced by quite reputable for top 10 manufacturers.

Limitation is part of the discussion section, not the conclusion section.

Conclusions

There is an overstated conclusion.

Page 11, lines 361-362: Given the absence of clear clinical or economic justification, and the potential for such 361 switching to undermine patient trust and adherence.

Author Response

Reviewer3 comment

-The author should explain the reason for choosing a hypertension or diabetes diagnosis.

Response

Thank you for this important question. We have added the rationale for selecting hypertension and diabetes in the Methods section, citing their high prevalence and chronic nature as appropriate for examining MSPS. We have also acknowledged this selection as a limitation in the manuscript.

Revised text:

We selected hypertension and diabetes for this study because both are highly prevalent in older Korean adults, are commonly managed with multi-source chronic medications, and entail frequent prescribing—making them ideal for investigating multi-source prescription switching (MSPS).

comment

Is there any national policy of switching between originator and generic drugs in Korea? Is there any national policy for physicians to only write generic drugs? Is there any policy to guarantee the quality of generic drugs? Is there any ‘me too’ drugs in Korea? Is there any policy for the maximum price of drugs in the market?

Response to the comment

We appreciate the reviewer’s attention to policy context. The Introduction we believe describes Korea’s regulatory framework: the legal allowance for generic substitution at dispensing, the uniform pricing policy, and MFDS oversight of bioequivalence to ensure generic quality. We also clarify that there is no national mandate for INN-based prescribing in Korea. We have avoided discussion of “me-too” drugs as they are beyond the study’s scope.

comment -The author should add any theory to explain the phenomenon, such as the theory of planned behavior (TPB). This study’s objective is physician behavior. Many factors affect drug prices, not only physician behavior, but also payer, availability (logistic), and regulation (Lee, 2020; Zarei, 2023).

Response

We appreciate the reviewer’s suggestion regarding theoretical framing. We have added a paragraph in the Discussion noting that physician-initiated MSPS can be partially understood through the lens of the Theory of Planned Behavior (TPB), which emphasizes the roles of attitudes, subjective norms, and perceived behavioral control in shaping clinical decision-making. This theoretical context helps explain how physician beliefs about generic equivalence, institutional norms, and system-level constraints (e.g., reimbursement policy, formulary structure) influence switching behavior.

Revised text

From a behavioral perspective, the patterns of MSPS observed in this study can be interpreted through the Theory of Planned Behavior (TPB), which posits that individual actions are shaped by three main constructs: attitudes, subjective norms, and perceived behavioral control (Ajzen, 1991). In the context of physician prescribing, attitudes may reflect skepticism toward the clinical equivalence of generics, despite their regulatory approval. Subjective norms could stem from the influence of peers, institutional culture, or pharmaceutical marketing, which subtly shape prescribing behaviors. Lastly, perceived behavioral control may be affected by institutional formularies, supply constraints, or ambiguity in national policy regarding generic substitution. Together, these factors may explain the discretionary and variable nature of MSPS, reinforcing the need for policies that not only address economic incentives but also behavioral drivers (Lee et al., 2020; Zarei et al., 2023).

Methods

comment -Did the author do factor analysis? What is the dominant factor affects the switching drug behavior?

Response

Thank you for your insightful question. We did not perform a formal factor analysis in this study, as our primary aim was descriptive—to quantify the extent of multi-source prescription switching (MSPS) and explore its variation across observable characteristics. Consequently, we did not statistically identify a single dominant factor. However, our stratified analysis suggests that institutional and drug market characteristics exert a greater influence on switching behavior than patient demographics, which may guide future hypothesis-driven research employing multivariate or latent variable modeling.

Results

comment -In Table 1: What does the author mean by ‘Negligible’ for ‘Price Differential between Brands & Generics’?

Response

Thank you for your question. By “Negligible,” we refer to the fact that, under Korea’s uniform pricing policy, there is no meaningful price difference between originator and generic drugs. The National Health Insurance (NHI) reimburses all pharmaceutically equivalent multi-source drugs—regardless of brand status—at the same fixed price. As a result, patients and providers have minimal financial incentive to prefer generics over originators based on price, reinforcing the absence of a price-based substitution dynamic in Korea's pharmaceutical market.

We have also replaced the term “brands” with “originators” to avoid confusion, as many generic products in Korea are marketed under proprietary brand names—i.e., branded generics.

comment -In Table 2: What is ‘N. of Sources’?

Response

Thank you for pointing this out. “N. of Sources” refers to the number of distinct bioequivalent drug products (i.e., manufacturers) available in the market for a same active ingredient, dose and formulation at the time of cohort entry. We replaced it with “Number of Sources.”

comment -This study evaluates switching between originator and generic drugs. Is it only for the same or for different active substances, too?

Response

Thank you for this important clarification. In this study, multi-source prescription switching (MSPS) refers strictly to switches between products that share the same active ingredient, dose and formulation, i.e., within pharmaceutically equivalent and bioequivalent drug products. We did not assess switching between drugs with different active substances. To avoid further confusion, we have clarified this definition in both the Methods section and relevant table notes.

Please note that we defined MSPS as below in the introduction section.

“This study aims to examine the occurrence of multi-source prescription switching (MSPS)—defined as physician-initiated switching between bioequivalent drug products—within ongoing drug therapy courses involving pharmaceutically equivalent medications.”

Discussion

comment -The author should discuss Table 1 comprehensively. Information in Table 1 implies that no drug quality issue because the generic drug is produced by quite reputable for top 10 manufacturers.

Response

Thank you for the thoughtful observation. We agree that Table 1 provides important context regarding the reputability of generic manufacturers. The fact that the majority of generics are produced by Korea’s top-ranked pharmaceutical companies—many of which are long-established and introduced world-leading drugs into the Korean market through licensing agreements —supports the interpretation that drug quality concerns are unlikely to explain the low generic substitution rate.

Revised text

Korean generics, however, are marketed under proprietary brand names and are often produced by reputable domestic pharmaceutical companies—many of which have historically introduced world-leading drugs into the Korean market through licensing agreements.

comment -Limitation is part of the discussion section, not the conclusion section.

Response

Thank you for pointing this out. We have revised the manuscript to relocate the limitations section from the conclusion to the discussion, in accordance with standard academic practice.

Conclusions

comment -There is an overstated conclusion.

-Page 11, lines 361-362: Given the absence of clear clinical or economic justification, and the potential for such 361 switching to undermine patient trust and adherence.

Response

Thank you for the valuable feedback. We agree that the phrasing may have overstated the certainty of the impact. We have revised the sentence to read as below.

Revised text:

Given the absence of clear clinical or economic justification, and given that such switching could potentially influence patient trust and adherence, MSPS may point to underlying inefficiencies in Korea’s pharmaceutical benefit design that warrant further exploration.
